# Genome-Wide Analysis of Differentially Expressed mRNAs and lncRNAs in Koi Carp Infected with Koi Herpesvirus

**DOI:** 10.3390/v14112555

**Published:** 2022-11-18

**Authors:** Zimin Yang, Wei Luo, Zhihong Huang, Min Guo, Xiaochuan He, Zihan Fan, Qing Wang, Qiwei Qin, Min Yang, Xuezhu Lee

**Affiliations:** 1Joint Laboratory of Guangdong Province and Hong Kong Regions on Marine Bio Resource Conservation and Exploitation, College of Marine Sciences, South China Agricultural University, Guangzhou 510642, China; 2Pearl River Fisheries Research Institute, Chinese Academy of Fishery Sciences, Key Laboratory of Fishery Drug Development of Ministry of Agriculture, Key Laboratory of Aquatic Animal Immune Technology, Guangzhou 510380, China; 3Laboratory for Marine Biology and Biotechnology, Qingdao National Laboratory for Marine Science and Technology, Qingdao 266000, China

**Keywords:** carp, KHV, lncRNA

## Abstract

Long noncoding RNAs (lncRNAs) constitute an emerging group of ncRNAs that modulate gene expression at the transcriptional or translational level. Koi herpesvirus (KHV), also known as Cyprinus herpesvirus type 3 (CyHV-3) and characterized by high pathogenicity and high mortality, has caused substantial economic losses in the common carp and koi carp fisheries industry. In this work, we sequenced the lncRNA and mRNA of host koi carp infected with KHV. A total of 20,178 DEmRNAs were obtained, of which 5021 mRNAs were upregulated and 15,157 mRNAs were downregulated. Both KEGG pathways and GO terms were enriched in many important immune pathways. The KEGG analysis showed that DEGs were significantly enriched in many important immune pathways, such as apoptosis, NOD-like receptor signaling pathway, Jak-STAT signaling pathway, TNF signaling pathway, IL-17 signaling pathway, NF-kappa B signaling pathway, and so on. Furthermore, a total of 32,697 novel lncRNA transcripts were obtained from koi carp immune tissues; 9459 of these genes were differentially expressed. Through antisense, cis-acting, and trans-acting analyses, the target genes of differentially expressed lncRNAs (DElncRNAs) were predicted. Gene ontology (GO) annotation and Kyoto Encyclopedia of Genes and Genomes (KEGG) pathway analyses showed that the DElncRNA expression pattern was consistent with the differential mRNA expression pattern. The lncRNA–mRNA network analysis, which included many immune pathways, showed that after KHV infection, the expression of most lncRNAs and their target mRNAs were downregulated, suggesting that these lncRNAs engage in a positive regulatory relationship with their target mRNAs. Considering that many studies have shown that herpesviruses can escape the immune system by negatively regulating these immune pathways, we speculated that these lncRNAs play a significant role in KHV’s escape from host immunity. Furthermore, 10 immune-related genes and 20 lncRNAs were subsequently verified through RT–qPCR, to confirm the accuracy of the high-throughput sequencing results. In this study, we aimed to explore lncRNA functions in the immune response of lower vertebrates and provide a theoretical basis for the study of noncoding RNAs in teleosts. Therefore, exploring lncRNA expression in KHV-infected koi carp helped us better understand the biological role played by lncRNA-dependent pathways in aquaculture animal viral infection.

## 1. Introduction

Long noncoding RNAs (lncRNAs) constitute an emerging group of ncRNAs that can modulate gene expression at the transcriptional or translational level. LncRNAs, comprising a heterogeneous class of noncoding transcripts, are endogenous cellular RNAs that are longer than 200 nucleotides [1]. Their length is similar to that of mRNA but the coding potential (open reading frames (ORFs) > 30 amino acids) is absent or reduced. Initially thought to exhibit nonbiological functions, lncRNAs were regarded as “transcriptional noise”. However, in recent years, lncRNAs have collectively become a hot topic in research, because of the potential role they play in the regulation of several biological processes via transcriptional or posttranscriptional regulatory mechanisms [2]. Notably, dysregulation of lncRNAs may alter mRNA expression or induce phenotypic changes that can induce abnormal responses or otherwise affect cell processes, such as cell proliferation, differentiation, migration, and apoptosis, in animals [3,4].

With the complex paradigm of RNA-based gene regulation slowly being elucidated, an increasing body of evidence has indicated that lncRNAs play significant roles in regulating host–virus interactions during viral infection [5]. Viruses can hijack host lncRNAs and modulate host signaling through chromatin remodeling and transcriptional and posttranscriptional control, to meet their own needs for replication [6,7].

In addition, mounting evidence has revealed that many lncRNAs are differentially expressed during infection with severe acute respiratory syndrome coronavirus (SARS-CoV, MA15), enterovirus 71, vesicular stomatitis virus, influenza A virus in mice, Zika virus, rabies virus, and hepatitis C virus. Viral lncRNAs control the host immune response, by regulating the expression of network genes and viral proliferation [8,9,10,11,12].

The total number of lncRNAs has been estimated to be encoded by ~20,000 transcripts; however, to date, only ~200 lncRNAs have been characterized. Although lncRNAs have been demonstrated to be involved in immune system regulation in mammals [9], the results of mammalian species lncRNA research are not easily translated to aquaculture species, because of the low evolutionary conservation among lncRNAs across species [13]. Therefore, related functional studies with teleosts are rare.

Koi herpesvirus (KHV), also known as Cyprinus herpesvirus type 3 (CyHV-3), was first reported in Israel and the United States in 1998. To date, KHV has caused large economic losses in the common carp and koi carp fisheries industry because, at water temperatures ranging from 18 °C to 28 °C, this viral infection is characterized by high pathogenicity and high mortality. However, no effective treatment for the disease is currently available. KHV is a double-stranded DNA virus with a 295-kbp genome encoding 156 predicted genes [14,15]. The mature virions are spherical and have a diameter of approximately 170–230 nm. They are mainly composed of a capsule, cortex, capsid, and core. Very limited data are available explaining the role played by lncRNAs in KHV infection in carp. Although lncRNAs encoded by common carp have been reported in some studies [16,17], no studies have focused on the relationship between carp lncRNAs and KHV infection.

In this work, we sequenced lncRNAs and mRNAs in the host (koi carp) after infection with KHV. We aimed to explore lncRNA functions and the lncRNA-mRNA co-expression network in the immune response of lower vertebrates and provide a theoretical basis for the study of noncoding RNAs in teleosts. Specifically, exploring the expression of lncRNAs and the targeting of lncRNA to mRNA in KHV-infected koi carp may help us better understand the biological role played by lncRNA-dependent pathways in aquaculture animal virus infection.

## 2. Materials and Methods

### 2.1. Sample Collection and Preparation

Koi fish weighing 158.13 ± 2.5 g were purchased from Huadiwan Market in Guangzhou City, Guangdong Province, China. After adapting to a circulating water system and being fed commercial feed every day for two weeks, the fish were randomly allocated to separate groups: an infected and uninfected group. The infected group was administered an intraperitoneal injection of 100 μL of CCB cell culture supernatant containing 100 PFU of KHV virus. The other group was the mock control and received an intraperitoneal injection of 100 μL of uninfected CCB cell culture supernatant, to eliminate the influence of the cell culture medium on the immune response. Three days after injection, we randomly selected uninfected and infected carp, and collected brain, gill, liver, spleen, and kidney tissue samples. Then, the tissues were mixed together and considered one fish sample. A total of three infected samples and three mock samples were ground with liquid nitrogen and incubated at −80 °C for the next step.

### 2.2. RNA Extraction, Library Construction, and Sequencing

A TRIzol reagent kit (Invitrogen, Carlsbad, CA, USA) was used for total RNA extraction, according to the manufacturer’s protocol. RNA quality was assessed with an Agilent 2100 Bioanalyzer (Agilent Technologies, Palo Alto, CA, USA) and verified using RNase free agarose gel electrophoresis. After total RNA was extracted, ribosomal RNAs (rRNAs) were removed to obtain a sample of mRNAs and ncRNAs. The enriched mRNAs and ncRNAs were fragmented into short sequences in fragmentation buffer and reverse transcribed into cDNA with random primers. Second-strand cDNA was synthesized using DNA polymerase I, RNase H, d NTP (dUTP, not dTTP), and buffer. Next, the cDNA fragments were purified with a QiaQuick PCR extraction kit (Qiagen, Venlo, The Netherlands) and end repaired. Poly (A) was added to the sequence, which was then ligated to Illumina sequencing adapters. Then, uracil-N-glycosylase (UNG) was used to digest second-strand cDNA. The digested products were size-selected using agarose gel electrophoresis, amplified by PCR, and sequenced with an Illumina Nova Seq6000 from Gene Denovo Biotechnology Co., Ltd. (Guangzhou, China).

### 2.3. Identification, Annotation, Prediction, and Structural Analysis of lncRNAs

Fastp (version 0.18.0) was used to filter the samples, to ensure high quality and clean reads. Parameters were selected to remove those reads that included adapter sequences or more than 10% unknown nucleotides, consisted of more than 50% low quality (Q-value ≤ 20) bases, or for which short read alignment was required. Bowtie2 (version 2.2.8) was used for mapping reads to the ribosome RNA database. The rRNA mapped reads were then removed. The remaining reads were used to assemble and analyze the transcriptome. Two software programs, CNCI [18] (version 2) and CPC [19] (version 0.9-r2) (http://cpc.cbi.pku.edu.cn/, accessed on 1 March 2022), were used to assess the protein-coding potential of novel transcripts, on the basis of default parameters. Reads at the intersection of nonprotein-coding potential results were chosen as long noncoding RNAs. Based on this analysis, lncRNAs were categorized into five classes, according to their location relative to protein-coding genes: intergenic lncRNAs, bidirectional lncRNAs, intronic lncRNAs, antisense lncRNAs, and sense overlapping lncRNAs.

### 2.4. Differential Expression and Functional Enrichment Analyses

The differentially expressed transcripts of coding RNAs and lncRNAs were analyzed between two different groups using DESeq2 [20] software and between two samples by edgeR [21]. Genes/transcripts with a false discovery rate (FDR) below 0.05 and absolute fold change ≥2 were considered differentially expressed genes (DEG)/transcripts. Differentially expressed coding RNAs were then subjected to GO functional term enrichment analysis (http://www.geneontology.org/, accessed on 22 March 2022) and KEGG pathway analysis [22].

### 2.5. LncRNA–mRNA Network Analysis

Pearson correlation coefficients of lncRNAs and mRNAs were calculated using the corresponding matrix data, and then their correlation was tested. The competing endogenous RNA (ceRNA) mechanism was found to be significantly and positively correlated with the co-expression of lncRNA–mRNA, with the correlation coefficient *r* > 0.95 and *P* < 0.001.

The lncRNA–mRNA targeted pairs were constructed based on the targeting relationship between differentially expressed mRNAs (DEmRNAs) and DElncRNAs. First, six pathways involved in KHV infection, including apoptosis, TNFα, NOD-like, IFN, NF-κB, and Toll-like pathways, were selected as analysis targets. A total of 50 lncRNAs and mRNAs with the most significant expression were screened, and an interaction network diagram was generated.

A ceRNA regulatory network (lncRNA–mRNA) was constructed based on the co-targeted lncRNA–mRNA pairs and co-expressed lncRNA–mRNA pairs. Then, the ceRNA regulatory network was visualized with Cytoscape software (https://cytoscape.org/, accessed on 28 April 2022).

### 2.6. Validation of the RNA-Sequencing Results via Quantitative PCR

To validate the accuracy of the RNA-sequencing results, a total of ten DEmRNAs and 20 DElncRNAs were selected, and their expression was quantified by reverse transcription-quantitative PCR. The gene-specific primers used for these selected lncRNAs and mRNAs are shown in Table 1. The Cyprinus carpio 18 s gene (Accession No. XR_006156360) was the reference gene. The cDNA was reverse transcribed with a ReverTra Ace qPCR RT Kit (TOYOBO, Kyoto, Japan). Fivefold-diluted cDNA and SYBR^®^ Green Real-time PCR Master Mix (TOYOBO, Kyoto, Japan) were used to perform qRT–PCR with Quant Studio 5 (Thermo ABI, San Diego, USA). Real-time PCR was performed in accordance with the ABI PRISM protocol, with a 50 µL reaction volume, consisting of 5 µL of template, 25 µL of SYBR^®^ Green Realtime PCR Master Mix, 2 µL of each primer (10 µM), and 16 µL of ddH2O. The cycling conditions were as follows: 1 min at 95 °C, followed by 40 cycles at 95 °C for 15 s, 60 °C for 15 s, and 72 °C for 45 s.

The comparative CT method for relative quantification was used to calculate the relative gene expression level [23]. Statistical analysis of the data at different stages obtained with the IBM SPSS Statistics 20 program was performed using a t test, and the difference in expression was considered to be significant when *p* < 0.05.

## 3. Results

### 3.1. Quality and Validation of Transcriptomic Data

Transcriptomic data sequencing was performed with an Illumina NovaSeq 6500 platform. The sequencing of each sample generated more than 12 billion reads. After filtering adapters, low-quality reads and poly (A) reads, high-quality clean reads, accounting for more than 99% of the raw reads, were obtained (Appendix A). Then, the cleaned reads were mapped to the reference genome, and more than 87% of these reads matched perfectly (Appendix A). To validate the accuracy of the sequencing data, ten DEmRNAs and 20 lncRNAs were randomly selected and quantified using RT–qPCR (Table 2 and Appendix A); these results showed trends similar to those of the RNA sequencing.

### 3.2. Identification of lncRNAs in Koi Carp Tissues

Based on the location of the transcripts in the reference genome, the transcripts were screened with CPC2 and CNCI software. Transcripts with a length ≥200 bp and exon number ≥2 were retained. In total, 32,697 novel lncRNA transcripts (Appendix A) were obtained from koi carp tissues, including 15,801 (48.32%) intergenic lncRNAs, 3033 (9.3%) sense lncRNAs, 918 (2.8%) antisense lncRNAs, 443 (1.3%) intronic lncRNAs, 461 (1.41%) bidirectional lncRNAs, and (11.42%) 3734 lncRNAs of other types (Figure 1). In addition, a total of 51,067 expressed genes, including 48,277 reference genes and 7272 novel genes, were detected in the transcriptomic data of the six samples (Appendix A).

### 3.3. Profiles of DEmRNAs and DElncRNAs

The number of mRNAs and lncRNAs shared between the infection group and the control group is shown in Figure 1. Based on the FDR < 0.05 and |log2FC| > 1 criteria, 20,178 mRNAs (Appendix A) and 9459 lncRNAs (Appendix A) were considered to be differentially expressed. Moreover, 5021 mRNAs were upregulated and 15,157 were downregulated in the treatment group (Figure 2A,B). Furthermore, 1336 upregulated and 8123 upregulated lncRNAs were identified (Figure 2C,D). DEmRNAs and DElncRNAs were used for cluster analysis. A heatmap indicated that the control and KHV-infected samples bifurcated into two separate clusters (Figure 2E, F). Therefore, two different groups of mRNAs and lncRNAs were expressed in patterns, indicating that the DEGs in the common carp tissues infected with KHV were significantly different from those in the control group.

### 3.4. GO Analyses of the DEGs

GO enrichment analysis was performed, to further investigate the potential functions of DEmRNAs in common crap tissues in response to KHV infection. The results of the GO enrichment analysis of differentially expressed mRNAs indicated 5359 functional groups (*P* <0.05, Appendix A), including 4064 groups in the biological process (BP), 421 groups in the cellular component (CC), and 874 groups in the molecular function (MF) categories. Directed acyclic graphs (DAGs) and histograms of GO functional annotation analysis of differentially expressed mRNAs showed that the BP category DEGs mRNAs were mainly enriched in primary metabolic processes, while the CC category mRNAs were enriched mainly in the cytoplasmic part and cytoplasm, and the most significant MF category mRNAs were enriched in RNA binding and electron carrier activity (Figure 3A,C,E). The 20 GO terms most enriched with mRNAs in the three ontologies categories (BP CC, and MF) are presented in Figure 3B,D and F.

### 3.5. KEGG Analyses of the DEGs

KEGG pathway analysis was performed based on the KEGG database “KEGG mapping tools for uncovering hidden features in biological data. Available online: http://www.genome.jp/kegg/pathway.html (accessed on 20 May 2022)”. A total of 341 KEGG pathways were found to be enriched with DEGs (Appendix A). Among the 20 KEGG pathways most enriched (Figure 4A), most were closely related to the immune response, including apoptosis, NOD-like receptor signaling pathway, Jak-STAT signaling pathway, TNF signaling pathway, IL-17 signaling pathway, NF-kappa B signaling pathway, C-type lectin receptor signaling pathway, Toll-like receptor signaling pathway, MAPK signaling pathway, Kaposi sarcoma-associated herpesvirus infection pathway, Epstein–Barr virus infection, T-cell receptor signaling pathway, cytokine–cytokine receptor interaction pathway, and lysosome pathway. In these signaling pathways, most of the genes that play an antiviral role in the immune response were significantly downregulated in the KHV-infected group; these included p53 and Apaf-1 in the apoptosis pathway; NOD1, NOD2, IL-1β, IL-6 and IL-8 in the NOD-like receptor signaling pathway; Jak and STAT1 in the Jak-STAT signaling pathway; TNF and TNFR and TRADD in the TNF signaling pathway; IL-17RA and IL-17RC in the IL-17 signaling pathway; p65, p100, TRAF6, and MyD88 in the NF-kappa B signaling pathway; PLCY2 in the C-type lectin receptor signaling pathway; TLR 2, TLR4, TLR7, TLR8, TLR9, CD80, and CD86 in the Toll-like receptor signaling pathway; CD28, CD38, CD45, and CD48 in the T-cell receptor signaling pathway; and CCL1, CCL3, CCL8, CCR2, CCR3, CCR4, CCR5, CCR7, and CCR8 cytokine receptor interaction (Appendix A).

Gene set enrichment analysis (GSEA) was performed using GSEA v4.0.2 software and the Molecular Signatures Database (Broad Institute). This analysis revealed several significantly enriched gene sets that regulated the antiviral immune response, including the MAPK signaling pathway, NF-kappa B signaling pathway, p53 signaling pathway, apoptosis, NOD-like receptor signaling pathway, and RIG-I-like receptor signaling pathway (Figure 4B). Moreover, the expression of most of the genes in these gene sets was downregulated, which was in accordance with the results of the KEGG analysis (Figure 4A).

### 3.6. Prediction and Functional Analysis of mRNA Targets of DElncRNAs

To investigate the potential regulatory role played by lncRNAs in koi carp tissues, we used GO and KEGG analyses to predict lncRNA target gene functions. We predicted the target genes of lncRNAs via antisense analysis. We also performed colocalization (cis-acting) and expression correlation (trans-acting) analysis of lncRNAs and protein-coding genes. The results of the GO term-enriched target gene predictions obtained via these three methods were almost identical; the BP category genes were mainly enriched in cellular processes and single-organism processes, while the CC category genes were mainly enriched in cells and cell parts. The most significant MF category-enriched genes involved binding and catalytic activity (Figure 5A–C). The KEGG enrichment results using the three methods were quite different. KEGG pathway enrichment analysis of target genes obtained by the antisense analysis revealed several immune pathways: Kaposi sarcoma-associated herpesvirus infection pathway, inflammatory bowel disease pathway, Vibrio cholerae infection pathway, epithelial cell signaling in the Helicobacter pylori infection pathway, hepatitis C pathway, and AMPK signaling pathway. The results of the KEGG enrichment analysis of the target genes predicted by the cis-acting method revealed additional immune signaling pathways, including the B-cell receptor signaling pathway, NF-kappa B signaling pathway, FoxO signaling pathway, Epstein–Barr virus infection, viral myocarditis, chemokine signaling pathway, and MAPK signaling pathway. When target genes obtained via the trans-acting method were analyzed, the KEGG results revealed pathways that play a significant regulatory role in antiviral immune response: apoptosis, NOD-like receptor signaling pathway, Jak-STAT signaling pathway, TNF signaling pathway, IL-17 signaling pathway, NF-Kappa B signaling pathway, C-type lection receptor signaling pathway, Toll-like receptor signaling pathway, MAPK signaling pathway, Kaposi sarcoma-associated herpesvirus infection, Epstein–Barr virus infection, T-cell receptor signaling pathway, cytokine–cytokine receptor interaction, and lysosome. Notably, these results were very strongly correlated with the KEGG enrichment results of differential mRNAs (Figure 5D–F). The detailed data about DElncRNAs targeting mRNAs are listed in Appendix A (one DElncRNA targeting one mRNA).

To highlight the function and relationship of DElncRNAs and mRNAs in KHV-infected koi carp, putative interactive networks of the DElncRNAs and DEmRNAs in several immune pathways were established using Cytoscape (v3.6) (Figure 6); these pathways included the apoptosis pathway (Figure 6A), TNFα pathway (Figure 6B), NOD-like pathway (Figure 6C), IFN and NF-κB pathway ((Figure 6D), and Toll-like pathway (Figure 6E). The correlation coefficients of all lncRNA–mRNA pairs were higher than 0.95 (*p* < 0.004). The results showed that only a few lncRNAs were associated with single-targeted mRNAs; for example, MSTRG.9580.5 targeted caspase 3(CAPS3) (Figure 6A), MSTRG.83593.5 targeted IL8 (Figure 6D), and MSTRG.9339.1 targeted RELA(Figure 6E); most DE targeted multiple genes in these signaling pathways; for example, the lncRNA MSTRG.1055982 targeted RIPKI, DAXX, BAD, Bax, Caspase 3 (CASP3), Caspase 6 (CASP6), Caspase 7 (CASP7), and Caspase 8 (CASP8) (Figure 6A); MSTRG.103187.2 targeted TLR4, MyD88, TNFRSF5, TRAF6, PIK3R1, PIK3CA, IL8, and CD80 (Figure 6E). In addition, some immune genes also formed relationships with multiple lncRNAs, such as the immune gene MyD88 in the TNFa signaling pathway, which was associated with multiple lncRNAs, including MSTRG.105598.2, MSTRG.90051.1, MSTRG.84010.1, MSTRG.104541.4, MSTRG.17435.1, MSTRG.84012.1, MSTRG.113937.3, MSTRG.103187.2, MSTRG.14010.1, MSTRG.40341.2, MSTRG.83783.1, MSTRG.837.2, MSTRG.117668.2, MSTRG.33048.1, MSTRG.65695.2, and MSTRG.99139.1. These results suggest that the effect of lncRNAs on these important immune signaling pathways was mediated through complex network patterns, not a single-pair regulation pattern. In addition, we found that most of the mRNAs in these immune pathways were downregulated and that most of the lncRNAs that targeted them were also downregulated. These findings indicated that most lncRNA–mRNA pairs were positively correlated, suggesting that these lncRNAs played a positive role in regulating the expression of their targeted mRNAs.

## 4. Discussion

A total of 20,178 DEmRNAs were obtained from the transcriptome database of common carp tissues. Of the mRNAs in the treatment group, 5021 mRNAs were upregulated, and 15,157 mRNAs were downregulated. Downregulated genes accounted for more than 75% of all DEGs. Both KEGG pathways and GO terms were enriched in some important immune pathways. The KEGG analysis showed that DEGs were significantly enriched in many important immune pathways, such as apoptosis, NOD-like receptor signaling pathway, Jak-STAT signaling pathway, TNF signaling pathway, IL-17 signaling pathway, NF-kappa B signaling pathway, C-type lectin receptor signaling pathway, Toll-like receptor signaling pathway, MAPK signaling pathway, Kaposi sarcoma-associated herpesvirus infection pathway, Epstein–Barr virus infection, T-cell receptor signaling pathway, Cytokine–cytokine receptor interaction pathway, and lysosome pathway. Many critical genes involved in these immune signaling pathways were downregulated, including p53, NOD1, NOD2, IL-1β, IL-6, IL-8, Jak, STAT1, TNF and TNFR, TRADD, IL-17RA, IL-17R, p65, p100, TRAF6, MyD88, TLR7, TLR9, CD80, CD38, CD45 aCD48, CCL3, CCL8, CCR3, CCR4, and CCR8. The GSEA results were compared with KEGG analysis results, which showed that most of the DEGs enriched in the MAPK signaling pathway, NF-Kappa B signaling pathway, p53 signaling pathway, apoptosis, NOD-like receptor signaling pathway, and Rig-I-like receptor signaling pathway were downregulated.

These pathways play important roles in the innate immune response of fish host cells, to defend against viral infection. When viruses invade host cells, viral pathogen-associated molecular patterns (PAMPs) are recognized by host cell pattern recognition receptors (PRRs), primarily NOD-like receptors (NLRs), Toll-like receptors (TLRs), and RIG-I-like receptors (RLRs). Subsequently, PRRs stimulate signaling that triggers an innate immune response, inducing the NF-kappa B signaling pathway, Jak-STAT signaling pathway, cytokine–cytokine receptor interaction pathway, and TNF signaling pathway. Activation of these signaling pathways leads to the production of cytokines, such as interferon, interferon-related genes, inflammatory factors, and chemokines, which either inhibit viral replication directly, promote immune cell migration, or induce apoptosis, thereby inhibiting viral replication [24,25,26].

In the studied herpesviruses, especially Kaposi’s sarcoma-associated herpesvirus (KSHV), these signaling pathways play crucial roles. Proteins in Toll-like pathways and NOD-like pathways play significant roles during the life cycle of KSHV. KSHV can infect human plasmacytoid dendritic cells (pDCs), activate the expression of CD83 and CD86, and induced IFN-α secretion from pDCs. Further analysis showed that the induction of IFN-α is mediated through TLR9 signaling activation [27]. Moreover, TLR7/8 signaling induces KSHV reactivation and lytic replication and allows the virus to propagate and escape a dying cell, which ensures KSHV survival and persistence for the lifetime of the infected host [28]. NLR signaling is also important during KSHV infection. KSHV encodes a viral homolog of cellular NLRP1 named ORF63, which is capable of broad inhibition of NLR inflammasome responses, suggesting that modulation of NLR-mediated innate immunity is important for the life-long persistence of herpes viruses [29].

In addition, the RIG-I-like receptor signaling pathway is critical during KSHV reactivation. RIG-I and its adaptor protein MAVS, two important mitochondrial antiviral proteins in the RIG-I-like receptor signaling pathway, can sense KSHV infection and suppress its replication following primary infection and/or viral reactivation. In fact, depletion of MAVS or RIG-I before KSHV reactivation leads to an increased viral load and infectious virus production, and MAVS depletion in latent KSHV-infected B cells leads to increased viral gene transcription [30]. Signaling pathways activated by these proteins can lead to the secretion of type I interferon (IFN-I) and host inflammasome response.

LncRNAs are produced from the genome and defined as transcripts >200 nucleotides in length that lack protein-coding potential [31]. LncRNAs can be roughly categorized into five classes on the basis of their genomic location: sense lncRNA, antisense lncRNA, intragenic lncRNA, intergenic lncRNA, and bidirectional lncRNA classes. LncRNAs function by interacting with RNA, DNA, proteins, or a combination of these molecules. On the basis of their localization, sequence, and/or secondary structure, the function of lncRNAs can often be determined. LncRNA-mediated gene regulation involves a wide variety of mechanisms. LncRNAs may act as epigenetic regulators, transcription promoters, decoys to repress transcription, or scaffolds to interact with protein partners and form ribonucleoprotein complexes [32,33,34]. Based on the levels of gene expression, lncRNA-mediated gene expression may take place at the transcriptional and/or posttranscriptional levels. LncRNAs may be involved in direct transcription by interacting with transcriptional complexes or DNA elements, such as promoters, which are involved in transcription [35]. More broadly, lncRNAs may participate in the modulation of chromatin structures by recruiting chromatin-modifying enzymes, inducing the expression or repression of a large number of genes [36].

In an integrative analysis of transcriptomic data, target genes of lncRNAs were predicted, mainly based on their functions: cis, antisense, or trans. Playing a cis role, the function of lncRNAs is related to adjacent protein-coding genes. LncRNAs located upstream and downstream of a protein-coding gene may regulate the expression of that gene. Playing an antisense role, similar to that of a microRNA (miRNA), the regulatory effect of a lncRNA is often related to complementary base pairing. Some antisense lncRNAs may regulate gene silencing, transcription, and mRNA stability, by binding to the mRNA of the justice chain. Playing a trans role, a lncRNA is related to co-expressed protein-coding genes. When lncRNAs are positively or negatively correlated with the expression levels of a certain gene, the target gene can be predicted via correlation analysis or co-expression analysis, using lncRNAs and protein-coding genes in two or more samples [37,38].

LncRNAs play critical roles in various biological processes and diseases, from cell differentiation and tissue organ development, to cancer metastasis. Many studies in mammals have shown that lncRNAs play important roles in the regulation of the immune response; for example, the TLR-stimulated lncRNA lincRNA-Cox2 can positively and negatively regulate the expression of distinct types of innate immune genes. Knocking down the long intergenic noncoding RNA (lincRNA) Cox2 disrupted the production of proinflammatory genes (i.e., IL-6), while IFN-related genes were hyperactivated in the absence of the lincRNA Cox2 [9]. In addition, emerging evidence has suggested that lncRNAs are involved in various pathways, such as p53, NF-κB, PI3K/AKT, and other signaling pathways [39]. The study of lncRNA functions in lower vertebrates is still in its infancy. Therefore, the functional roles played by lncRNAs in lower vertebrates remain unknown. In Miichthys miiuy, several lncRNAs, named MAVS antiviral-related lncRNA (MARL), IRAK4-related lncRNA (IRL), nucleotide oligomerization domain 1 (NOD1) antibacterial, and antiviral-related lncRNA (NARL), have been identified. A functional analysis showed that these fish lncRNAs play essential roles in host antiviral responses and inhibition of Siniperca chuatsi rhabdovirus (SCRV) replication [40,41,42]. These studies indicated that fish lncRNAs play important regulatory functions in various biological processes, similarly to those of mammals.

Furthermore, the lncRNA–mRNA network analysis of these immune pathways also showed that after KHV infection, the expression of most lncRNAs and their target mRNAs was downregulated, suggesting that these lncRNAs engage in a positive regulatory relationship with their target mRNAs. This network analysis will provide a reference for our future studies. Next, we will focus on verifying the targeted relationship of these lncRNA–mRNA pairs and discuss how these lncRNAs affect important immune pathways, to enable the immune escape of KHV. Moreover, as we previously inferred, KHV escapes from the host immune system by negatively regulating the lncRNA-associated immune pathways. Therefore, we believe that lncRNAs play a significant role in KHV escape from host immunity. KHV may not only regulate mRNA expression but also regulate lncRNA expression and even regulate important target mRNAs, by regulating lncRNA expression. In other words, KHV may directly regulate mRNA expression, on the one hand, and indirectly regulate mRNA through the regulation of lncRNA, on the other hand.

## 5. Conclusions

In this study, 32,697 novel lncRNA transcripts (Additional file S1) were obtained from koi carp tissues, and 9459 of these lncRNAs were differentially expressed: 1336 DElncRNAs were upregulated, and 8123 DElncRNAs were upregulated. The target genes of DElncRNAs were predicted through antisense, cis-acting, and trans-acting analyses. For target genes predicted using the trans-acting method, the KEGG enrichment result was consistent with the KEGG enrichment result of differentially expressed mRNAs. The main enriched pathways were apoptosis, the NOD-like receptor signaling pathway, Jak-STAT signaling pathway, TNF signaling pathway, IL-17 signaling pathway, NF-Kappa B signaling pathway, C-type lectin receptor signaling pathway, Toll-like receptor signaling pathway, the MAPK signaling pathway, Kaposi sarcoma-associated herpesvirus infection, Epstein–Barr virus infection, T-cell receptor signaling pathway, cytokine–cytokine receptor interactions, and lysosomes. The lncRNA–mRNA network analysis, which included many immune pathways, showed that the expression of most lncRNAs and their target mRNAs was downregulated. Considering that many studies have shown that herpesviruses can escape the immune system by negatively regulating these immune pathways, we speculate that these lncRNAs play a significant role in KHV’s escape from host immunity. The regulatory mechanism of these lncRNAs will be the focus of our future research.

## Figures and Tables

**Figure 1 viruses-14-02555-f001:**
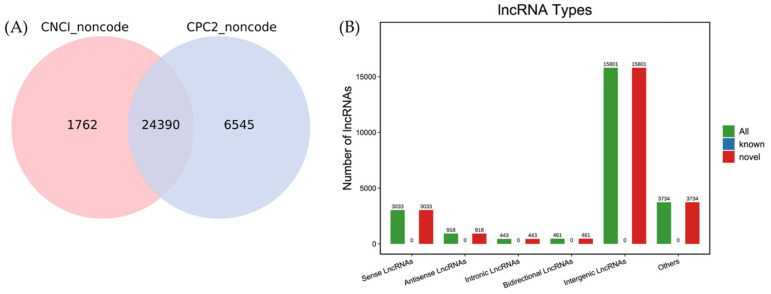
Screening and classification of lncRNAs in koi carp. (**A**) The number of LncRNAs predicted by CPC2 and CNCI. The CPC2 and CNCI databases were used to analyze the coding potential of novel lncRNAs. RNAs identified using two analytical tools were chosen as candidate lncRNAs. (**B**) The novel lncRNAs were classified mainly as sense lncRNAs, antisense lncRNAs, intronic lncRNAs, bidirectional lncRNAs, intergenic lncRNAs, and others.

**Figure 2 viruses-14-02555-f002:**
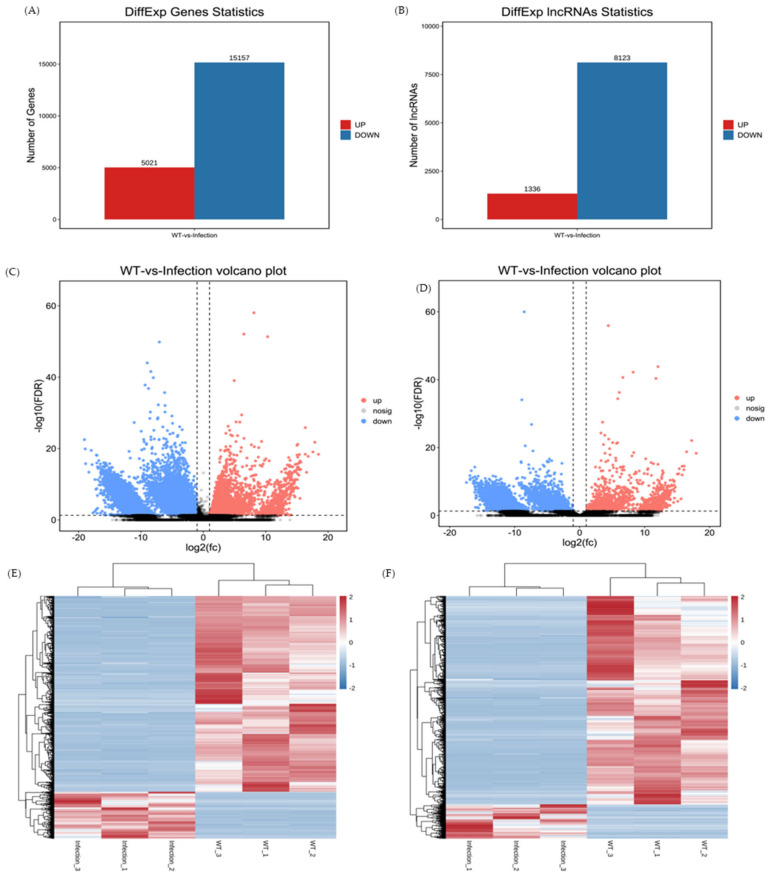
Differential expression of mRNAs and lncRNAs in the control and KHV-infected koi carp. Number of differentially expressed mRNAs (DEmRNAs) (**A**) and lncRNAs (**B**). Volcano plots showing differentially expressed mRNAs (**C**) and lncRNAs (**D**). Hierarchical clustering of the expression profiles of differentially expressed mRNAs (**E**) and lncRNAs (**F**).

**Figure 3 viruses-14-02555-f003:**
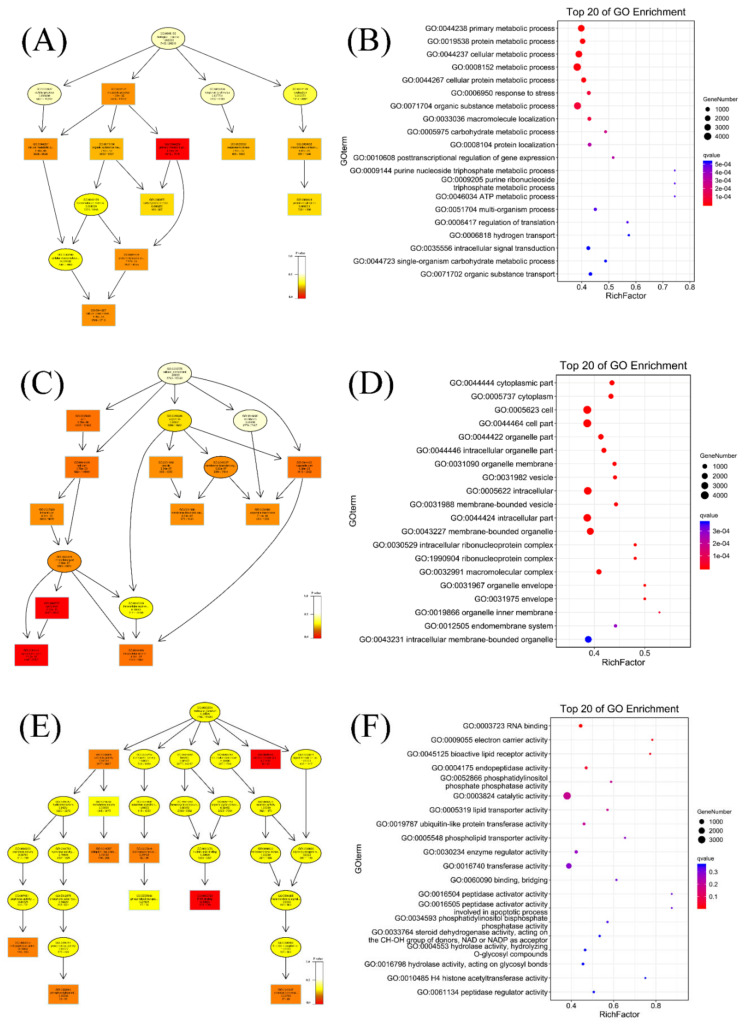
GO enrichment analysis of differentially expressed mRNAs (DEmRNAs). (**A**) The directed acyclic graph (DAG) of the BP category, (**B**) and the 20 most enriched GO terms in the BP category. (**C**) The DAG of the CC category, (**D**) and the 20 most enriched GO terms in the CC category. (**E**) The DAG of the MF category (**F**), and the 20 most enriched GO terms in the MF category.

**Figure 4 viruses-14-02555-f004:**
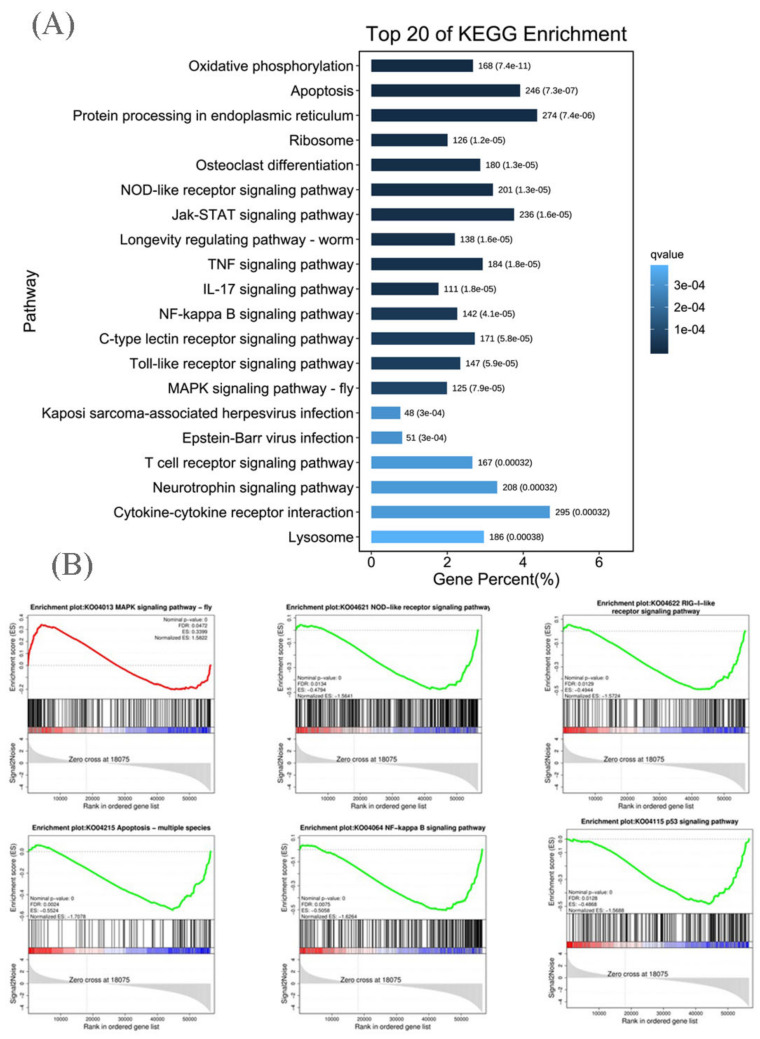
Pathway enrichment analysis of DEmRNAs. (**A**) The 20 most enriched KEGG pathways. (**B**) Gene-set enrichment analysis (GSEA).

**Figure 5 viruses-14-02555-f005:**
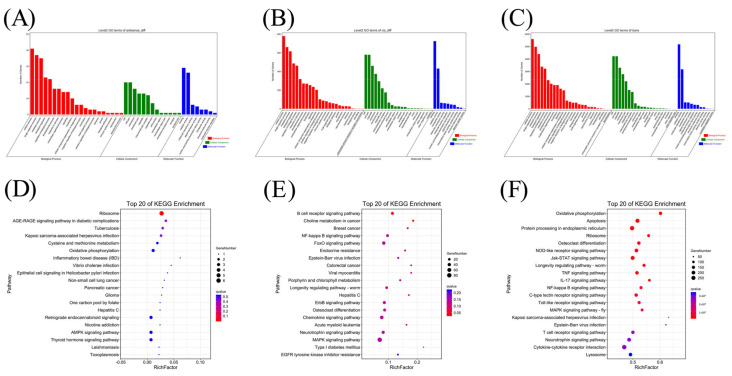
GO and KEGG enrichment analyses of potential target mRNAs of differentially expressed lncRNAs (DElncRNAs). (**A**) GO analysis of potential target mRNAs predicted using antisense analysis. (**B**) GO analysis of potential target mRNAs predicted using colocation (cis-acting) analysis. (**C**) GO analysis of potential target mRNAs predicted using expression correlation (trans-acting) analysis. (**D**) KEGG analysis of potential target mRNAs predicted using antisense analysis. (**E**) KEGG analysis of potential target mRNAs predicted using colocation (cis-acting) analysis. (**F**) KEGG analysis of potential target mRNAs predicted using expression correlation (trans-acting) analysis.

**Figure 6 viruses-14-02555-f006:**
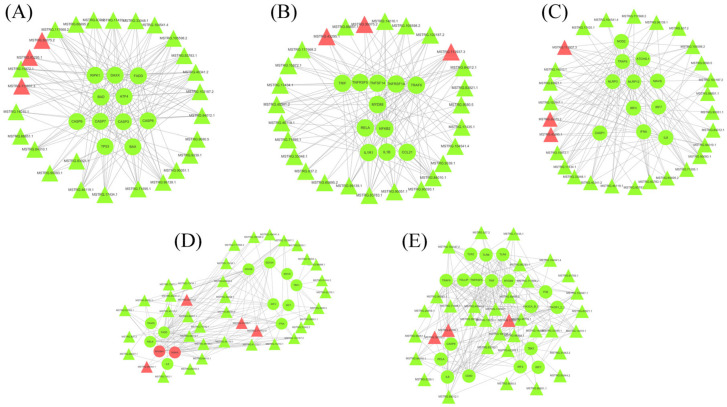
Co-expression network of differentially expressed lncRNAs (DElncRNAs) and mRNAs in KHV-infected koi carp. (**A**) Co-expression network of DElncRNAs and the predicted target mRNAs in the apoptosis pathway. (**B**) Co-expression network of DElncRNAs and the predicted target mRNAs in the TNFα pathway. (**C**) Co-expression network of DElncRNAs and their predicted target mRNAs in the NOD-like pathway. (**D**) Co-expression network of DElncRNAs and their predicted target mRNAs in the IFN and NF-κB pathways. (**E**) Co-expression network of DElncRNAs and their predicted target mRNAs in the Toll-like pathway. The circles are nodes that represent mRNAs, and the triangles are nodes that represent lncRNAs. The red nodes represent upregulated RNAs, and green nodes represent downregulated RNAs.

**Table 1 viruses-14-02555-t001:** Primers used for real-time PCR analysis. F: Forward primers; R: reverse primers.

Primer	Sequence (5′to 3′)
31942.2-qpcr-F	TGACTGGGAGTGATGGTTTA
31942.2-qpcr-R	AGAATGCCTTGAGGGTGAGT
84010.1-qpcr-F	CCACCTAGTGGTCAGGAGAT
84010.1-qpcr-R	GACCCACCGAATACAAAGAG
837.2-qpcr-F	CCCGAGAATCGTTCTGTAGTGG
837.2-qpcr-R	TCTGGGCCTAATGCTGTCAATC
84012.1-qpcr-F	TGACCCACCGAATACAAAGA
84012.1-qpcr-R	GAGGCTATAACTCCGCAACG
9580.5-qpcr-F	GGGGATTCTCCTTGCTGTAT
9580.5-qpcr-R	GGGCCATCTATCTTCTTTGC
4779.1-qpcr-F	TTTATTTAAGGGTGGCAGTG
4779.1-qpcr-R	TTCTTTGAGGGTTTGTTTCA
5436.4-qpcr-F	AACAACTGCCGCATAATGAAAC
5436.4-qpcr-R	AGCAACCATAGACCAGCAACAA
15250.1-qpcr-F	AAGTGAACTGTATGGCAGAG
15250.1-qpcr-R	ATGAAAGAATGAACGGATGA
40348.1-qpcr-F	ATTTGTTGGTGCTGTTGTCC
40348.1-qpcr-R	TGAGGCTTTACGCATAGGTT
40352.1-qpcr-F	ATTTGTTGGTGCTGTTGTCC
40352.1-qpcr-R	AGGCTTTACGCATAGGTTGT
44075.1-qpcr-F	CGCATTTCATAATATCCACC
44075.1-qpcr-R	ATAACACCAGAAGCAGACCA
52488.1-qpcr-F	GAGTCGTTGCTATCGCTTGG
52488.1-qpcr-R	ACCTCTGCCTCATCTTTCTC
52508.1-qpcr-F	AGGAGAAAGATGAGGCAGAG
52508.1-qpcr-R	TGAGGAGATTGAAGTGGAGG
64495.2-qpcr-F	CCACCTTGCGACTAAATTCTAAATC
64495.2-qpcr-R	TACAAAATTGAAAGCCAGTCACAAC
65064.1-qpcr-F	GCGGTGTTCTTCTGTCTGCA
65064.1-qpcr-R	GCGTCGGGTCTTGGGTTCTA
73364.3-qpcr-F	TGCTGGGAGGCTGAGGCTGG
73364.3-qpcr-R	CGCGGACACCCGATCGACAT
76443.1-qpcr-F	CCTTCGCCCTTGTTTCGGTG
76443.1-qpcr-R	AGACACGGGAGTCGGGAGGC
80990.1-qpcr-F	CCTCTTAATTGAATGCCCCTCCTC
80990.1-qpcr-R	TCTCCGTCATAGCGGTTTTACTTG
87639.1-qpcr-F	TCAGTAAATTCGATTTTAATGTGGG
87639.1-qpcr-R	ATTGAAGTATTTTCTATCCTGTGGC
91318.1-qpcr-F	GGGGCAACAGCTTCTAATGT
91318.1-qpcr-R	CCAGCCACCAAAGACAGATT
BAD-QPCR-F	ATGTGCGTGGAAAGCGTCAAC
BAD-QPCR-R	AAAGGCTCCGATGGTCACTCC
Bax-qpcr-F	GGCTATTTCAACCAGGGTTCC
Bax-qpcr-R	TGCGAATCACCAATGCTGT
MYD88-QPCR-F	AGCCTTTGCCCAGGAACTC
MYD88-QPCR-R	TGTGTGGAGGGTCTGGTGTA
TRAF6-QPCR-F	CCGGACCGAAACAGTATAATGGC
TRAF6-QPCR-R	CCACGTCATAGCCTTGCTGA
IRF-QPCR-3F	CAGGCATACGGAGGACATT
IRF-QPCR-3R	TGGCTTCAGGTCTGTTTTTG
ATG16L-qpcr-F	ATCAGGTCGGAGAGTATCGT
ATG16L-qpcr-R	GTTTTGTCTAGTTTCCCCGT
MAVS-QPCR-F	TCACACTCACTGATAGGGAAGAG
MAVS-QPCR-R	TAGCTCTCATCTCATTAGCCAGT
TBK1-QPCR-F	AGAGGAGTATCTGCATCCTGAC
TBK1-QPCR-R	CCTTCGTACGGTCTGAACG
RELA-QPCR-F	CACTGTAACTGTGTGTGTTTGTCT
RELA-QPCR-R	CCGCTGTAGTTGTGAACCTTGA
IFN-1-QPCR-F	ACCAGGTGAGGTTTCTTGTC
IFN-1-QPCR-R	CCACTGTCGTTAGGTTCCAT

**Table 2 viruses-14-02555-t002:** Verification of differential mRNA and lncRNA expression during KHV infection. The selected DEmRNAs and DElncRNAs were validated using quantitative real-time PCR. Fold changes were normalized against the 18S rRNA. “+” indicates upregulated; “-” indicates downregulated; “Y” indicates that the real-time PCR result was consistent with the high-throughput sequencing result.

Gene	Sequencing Result (Infection vs. NC)	Real-Time PCR Validation Result (Infection vs. NC)	Consistent or Not
837.2	-	-	Y
4779.1	-	-	Y
5436.4	-	-	Y
9580.5	-	-	Y
15250.1	-	-	Y
31942.2	-	-	Y
40348.1	-	-	Y
40352.1	-	-	Y
44075.1	-	-	Y
52488.1	-	-	Y
52508.1	-	-	Y
64495.2	-	-	Y
65064.1	-	-	Y
73364.3	+	+	Y
76443.1	-	-	Y
80990.1	-	-	Y
84010.1	-	-	Y
84012.1	-	-	Y
87639.1	-	-	Y
91318.1	-	-	Y
BAD	-	-	Y
Bax	-	-	Y
Myd88	-	-	Y
TRAF6	-	-	Y
IRF3	-	-	Y
ATG16L	-	-	Y
MAVS	-	-	Y
TBK1	-	-	Y
RELA	-	-	Y
IFN1	-	-	Y

## Data Availability

The datasets presented in this study can be found in online repositories. The names of the repository/repositories and accession number(s) can be found at GSA: https://mp.weixin.qq.com/s/PxuQnUGj7TFWqKW2EH9UXw with a project NO. PRJCA010550 (accessed on: 9 December 2021).

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
