# Peer review of "Genome-Wide Analysis of Differentially Expressed mRNAs and lncRNAs in Koi Carp Infected with Koi Herpesvirus"

_viruses, 2022, doi:10.3390/v14112555_

Round 1
Reviewer 1 Report
The study by Yang et al. shows data on “Genome-wide analysis of differentially expressed mRNAs and 2 lncRNAs in koi carp infected with koi herpesvirus”. A total of 32,697 novel 25 lncRNA transcripts were obtained from koi carp immune tissues; 9,459 of these genes were differentially expressed during KHV infection. GO annotation and KEGG pathway analyses showed that the DElncRNA expression pattern was consistent with that of the differential mRNA expression; DElncRNA target genes were enriched in several pathways related to immunity, such as apoptosis, NOD-like receptor signaling pathway, Jak-STAT signaling pathway, TNF signaling pathway, IL-17 signaling pathway, NF-Kappa B signaling pathway, C-type ligand–receptor signaling pathway, Toll-like receptor signaling pathway, and MAPK signaling pathway. The manuscript could be improved by addressing the comments listed below:
1. We cannot read the partial figures clearly in viruses-1992092-peer-review-v1, the quality of Fig.3, Fig.5 and Fig.6 should be improved. In this manuscript, we just got so many lncRNAs (9,459) were differentially expressed and these lncRNAs involve with the regarding immune pathways during KHV infection.
2. In the part “Prediction and functional analysis of mRNA targets of DElncRNAs by GO and KEGG enrichment analyses” there is no the detail data about DElncRNAs targeting mRNAs (one DElncRNA targeting one mRNA).
3. The discussion part (Line 364-493) fucus on lengthy review of research progress and general concept about innate immune pathways,but lack of the detailed discussion about the results of this manuscript.
Author Response
Following are the revisions and responses according to referee’s comments.
Reviewer #1: The study by Yang et al. shows data on “Genome-wide analysis of differentially expressed mRNAs and 2 lncRNAs in koi carp infected with koi herpesvirus”. A total of 32,697 novel 25 lncRNA transcripts were obtained from koi carp immune tissues; 9,459 of these genes were differentially expressed during KHV infection. GO annotation and KEGG pathway analyses showed that the DElncRNA expression pattern was consistent with that of the differential mRNA expression; DElncRNA target genes were enriched in several pathways related to immunity, such as apoptosis, NOD-like receptor signaling pathway, Jak-STAT signaling pathway, TNF signaling pathway, IL-17 signaling pathway, NF-Kappa B signaling pathway, C-type ligand–receptor signaling pathway, Toll-like receptor signaling pathway, and MAPK signaling pathway. The manuscript could be improved by addressing the comments listed below:
- We cannot read the partial figures clearly in viruses-1992092-peer-review-v1, the quality of Fig.3, Fig.5 and Fig.6 should be improved. In this manuscript, we just got so many lncRNAs (9,459) were differentially expressed and these lncRNAs involve with the regarding immune pathways during KHV infection.
Response: Thanks for your carefully checking and professional advice, we replaced the Fig.3, Fig.5 and Fig.6 with high quality ones to ensure that the quality of the figures were suit for publication.
- In the part “Prediction and functional analysis of mRNA targets of DElncRNAs by GO and KEGG enrichment analyses” there is no the detail data about DElncRNAs targeting mRNAs (one DElncRNA targeting one mRNA).
Response: Thank you for your carefully checking, this suggestion is very useful. Indeed, this part of data should be presented as part of our article. Therefore, we accepted this suggestion, and the data about DElncRNAs targeting mRNAs were sorted out as additional file 6, which was attached below our article.
- The discussion part (Line 364-493) fucus on lengthy review of research progress and general concept about innate immune pathways,but lack of the detailed discussion about the results of this manuscript.
Response: Done. Thanks for your carefully reviewing. We accept this suggestion, to streamline this section, we have removed paragraphs that are less relevant to the results (Line 364-493 of the original manuscript).

Reviewer 2 Report
The present study provided a transcriptomic analysis of mRNAs and lncRNAs in koi herpesvirus infected koi carp.
1. Large part of the results and discussion are related to mRNAs. But the part related to mRNA is lacked in Abstract and Introduction.
2. The tissues of brain, gill, liver, spleen, and kidney were collected for the transcriptome analysis. The authors stated the tissues as immune tissues in results and discussion. Are they all immune tissues?
3. The time that collect the tissues should be described.
4. Table 2 is too simple. What is the product of these genes, and the fold changes in transcriptome and RT-qPCR?
5. Fig3, what the different color in (A) (C)(E) means?
6. The Discussion part is lengthiness. I think the theme of this MS is not the detailed description of these immune related pathway or genes that have been proved.
7. A functional verification about the mRNA-lncRNA-virus infection is lacked, which made the MS just a summary of the informative data.
8. The authors stated that 201718 mRNAs were considered to be DEGs. I don’t know common carp has so much different mRNAs.
Author Response
Following are the revisions and responses according to referee’s comments.
Reviewer #2:
- Large part of the results and discussion are related to mRNAs. But the part related to mRNA is lacked in Abstract and Introduction.
Response:Thanks for your professional advising and reviewing, we accept this suggestion, the mRNA part of abstract and introduction were added in Line 25-30, Line 36-42 for abstract and Line 97-101 for introduction part, respectively.
- The tissues of brain, gill, liver, spleen, and kidney were collected for the transcriptome analysis. The authors stated the tissues as immune tissues in results and discussion. Are they all immune tissues?
Response:Thanks for your carefully reviewing and kindly revise. It is true that brain is not immune tissue, although it is the main target tissue of KHV, we have revised the statement of immune tissue in the whole article to improve the accuracy of our description.
- The time that collect the tissues should be described.
Response:Done. The time of collecting were shown in Line 113.
- Table 2 is too simple. What is the product of these genes, and the fold changes in transcriptome and RT-qPCR?
Response:Done. These genes were selected from those DElncRNAs that shown the most significant different expression in transcriptome data. The RT-qPCR data in Table 2 were rearranged and shown in new Fig. S1, in which the fold changes could be seen clearly.
- Fig3, what the different color in (A) (C)(E) means?
Response:Thanks for your carefully reviewing and kindly revise. In Figure 3, the different colours represent the significance of the Go enrichment, with darker colours representing smaller p-values, and to illustrate this more clearly, we have inserted the following figure into Figure 3.
- The Discussion part is lengthiness. I think the theme of this MS is not the detailed description of these immune related pathway or genes that have been proved.
Response:Done. Thanks for your carefully reviewing. We accept this suggestion, in order to streamline this section, we have removed paragraphs that are less relevant to the results (Line 364-493 of the original manuscript).
- A functional verification about the mRNA-lncRNA-virus infection is lacked, which made the MS just a summary of the informative data.
Response:Thank you for your professional revise, the functional verification about the mRNA-lncRNA-virus infection is a key part of our research study in future. In fact, we have already found some DElncRNAs that might play a key role in regulating koi herpes virus replication via targeting mRNA. The rough experimental data were shown in this followed figure, this part of data was unpublished and still in study, please look forward to our related research work and the coming article.
- The authors stated that 201718 mRNAs were considered to be DEGs. I don’t know common carp has so much different mRNAs.
Response:Thanks for your carefully reviewing, this might due to a careless mistake, the correct number is 20178 and we have corrected it in the article Line 220 and Line 359.

Round 2
Reviewer 1 Report
In the resubmitted manuscript " Genome-wide analysis of differentially expressed mRNAs and 2 lncRNAs in koi carp infected with koi herpesvirus " Yang et al. Typos have been corrected and language has been improved. And the concern from the first submission has been replied.
1)The authors have replaced the Fig.3, Fig.5 and Fig.6 with high quality ones to ensure that the quality of the figures be suitable for publication.
2)The data about DElncRNAs targeting mRNAs have been attached as additional file 6 by the authors.
3)The previous comment “The discussion part (Line 364-493) fucus on lengthy review of research progress and general concept about innate immune pathways,but lack of the detailed discussion about the results of this manuscript.” The authors have removed paragraphs that are less relevant to the results (Line 364-493 of the original manuscript) and improved the discussion part. This is considered sufficiently, although it would be more optimal to discuss the results of this manuscript in detail.